# Comparison of the New Neo-Glasgow Prognostic Score Based on the Albumin-Bilirubin Grade with Currently Used Nutritional Indices for Prognostic Prediction following Surgical Resection of Hepatocellular Carcinoma: A Multicenter Retrospective Study in Japan

**DOI:** 10.3390/cancers14092091

**Published:** 2022-04-22

**Authors:** Masaki Kaibori, Atsushi Hiraoka, Hiroya Iida, Koji Komeda, Fumitoshi Hirokawa, Masaki Ueno, Hisashi Kosaka, Kosuke Matsui, Mitsugu Sekimoto

**Affiliations:** 1Department of Surgery, Kansai Medical University, Osaka 573-1191, Japan; kosakahi@hirakata.kmu.ac.jp (H.K.); matsuik@hirakata.kmu.ac.jp (K.M.); sekimotm@hirakata.kmu.ac.jp (M.S.); 2Gastroenterology Center, Ehime Prefectural Central Hospital, Matsuyama 790-0024, Japan; hirage@m.ehime-u.ac.jp; 3Department of Surgery, Shiga University of Medical Science, Otsu 520-2192, Japan; hiroya@belle.shiga-med.ac.jp; 4Department of General and Gastroenterological Surgery, Osaka Medical and Pharmaceutical University, Takatsuki 569-8686, Japan; koji.komeda@ompu.ac.jp (K.K.); fumitoshi.hirokawa@ompu.ac.jp (F.H.); 5Second Department of Surgery, Wakayama Medical University, Wakayama 641-8509, Japan; ma@wakayama-med.ac.jp

**Keywords:** hepatocellular carcinoma, albumin-bilirubin grade, Glasgow prognostic score, complication, neo-Glasgow prognostic score

## Abstract

**Simple Summary:**

Clinical usefulness of the recently developed neo-Glasgow prognostic score (GPS) as a nutritional prognostic assessment in hepatocellular carcinoma (HCC) were evaluated in a multicenter retrospective study. In multivariate analysis with the Cox proportional hazards model, elevated alpha-fetoprotein (AFP; ≥100 ng/mL; hazard ratio [HR] 2.190, *p* < 0.001), multiple tumors (HR 1.784, *p* = 0.006), tumor size of ≥5 cm (HR 1.508, *p* = 0.032), and neo-GPS of ≥1 (HR 1.554, *p* = 0.019) were significant prognostic factors for OS, whereas elevated AFP (≥100 ng/mL) (HR 1.743, *p* < 0.001), multiple tumors (HR 1.537, *p* = 0.004), and neo-GPS of ≥1 (HR 1.522, *p* = 0.001) were significant prognostic factors for PFS. A neo-GPS of ≥1 was associated with higher rate of high-grade (≥3) Clavien-Dindo complications than a neo-GPS of <1 (31.1% vs. 16.7%, *p* = 0.007). Neo-GPS was a good prognostic nutritional assessment tool for the prediction of postoperative complications and prognosis in patients undergoing surgical HCC resection.

**Abstract:**

Nutritional assessment is important for predicting a prognosis in hepatocellular carcinoma (HCC). The authors examined the utility of the recently developed neo-Glasgow prognostic score (GPS) as a nutritional prognostic assessment in HCC in a multicenter retrospective study of 271 patients with HCC and Child-Pugh class A liver function who underwent R0 resection between 2011 and 2013. The median age was 72 years, 229 and 42 patients had Child-Pugh scores of 5 and 6, respectively, 223 patients had single tumors, the median tumor size was 3.6 cm, and open and laparoscopic resection were performed in 138 and 133 patients, respectively. We compared the prognostic predictive utility of the prognostic nutritional index, neutrophil/lymphocyte and platelet/lymphocyte ratios, controlling nutritional status score, GPS, and neo-GPS, which uses albumin-bilirubin grade (ALBI) instead of albumin. The c-indexes for the predictive prognostic value for overall survival (OS) and progression-free survival (PFS) were best for neo-GPS (OS: 0.571 vs. ≤0.555; PFS: 0.555 vs. ≤0.546). In multivariate analysis with the Cox proportional hazards model, elevated alpha-fetoprotein (AFP; ≥100 ng/mL; hazard ratio [HR] 2.190, 95% confidence interval [CI] 1.493–3.211, *p* < 0.001), multiple tumors (HR 1.784, 95%CI 1.178–2.703, *p* = 0.006), tumor size of ≥5 cm (HR 1.508, 95%CI 1.037–2.193, *p* = 0.032), and neo-GPS of ≥1 (HR 1.554, 95%CI 1.074–2.247, *p* = 0.019) were significant prognostic factors for OS, whereas elevated AFP (≥100 ng/mL) (HR 1.743, 95%CI 1.325–2.292, *p* < 0.001), multiple tumors (HR 1.537, 95%CI 1.148–2.057, *p* = 0.004), and neo-GPS of ≥1 (HR 1.522, 95%CI 1.186–1.954, *p* = 0.001) were significant prognostic factors for PFS. A neo-GPS of ≥1 was associated with a higher rate of high-grade (≥3) Clavien-Dindo complications than a neo-GPS of <1 (31.1% vs. 17.0%, *p* = 0.007). Neo-GPS was a good prognostic nutritional assessment tool for the prediction of postoperative complications and prognosis in patients undergoing surgical HCC resection.

## 1. Introduction

Hepatocellular carcinoma (HCC) is the most common primary malignancy of the liver and the fifth most common malignancy overall [1]. Nutritional assessment is important for predicting prognosis in patients undergoing surgical treatment for malignant tumors, including those requiring surgical resection for HCC. The prognostic nutritional index (PNI) [2,3], neutrophil/lymphocyte ratio (NLR) [4], platelet/lymphocyte ratio (PLR) [5], and controlling nutritional status (CONUT) score [6] have been proposed as nutritional prognostic assessment tools. In addition, the Glasgow prognostic score (GPS) [7,8,9], which uses C-reactive protein (CRP) and serum albumin cutoff values of 1.0 mg/dL and 3.5 g/dL, respectively, has been shown as an important and useful nutritional assessment tool for predicting prognosis in patients with malignant tumors. For example, the rate of complications following surgical resection for advanced gastric cancer was reported to be higher in patients with a GPS of ≥1 compared with those with a lower GPS (31% vs. 16%, *p* = 0.022) [10]. The authors have recently introduced neo-GPS, which replaces albumin level with albumin-bilirubin grade [11], which can discern borderline amino acid imbalance [12], as a prediction tool for prognosis following surgical resection.

In the present study, we aimed to evaluate the prognostic predictive value of neo-GPS for predicting a prognosis following surgical resection in patients with HCC and to compare its utility with other currently used nutritional indices.

## 2. Materials and Methods

### 2.1. Study Design

The medical records of all patients with HCC who underwent liver resection in Kansai Medical University Hospital, Shiga University of Medical Science, Osaka Medical and Pharmaceutical University, and Wakayama Medical University Hospital between 2011 and 2013 were screened. During the study period, 271 patients with HCC and Child-Pugh class A liver function who underwent R0 resection, defined as the macroscopic removal of all tumor lesions, were enrolled in the present multicenter retrospective study. The study protocol was approved by the Institutional Ethics committee of Kansai Medical University (reference number: KMU 2021311).

### 2.2. Determination of Underlying Liver Disease

HCC was considered to be due to the hepatitis C virus (HCV) in cases with anti-HCV antibody positivity, and due to hepatitis B virus (HBV) in cases with HBV surface antigen positivity.

### 2.3. Assessment of Liver Function and Nutritional Status

Child-Pugh scores, classification [13], and ALBI grade [14,15] were used to assess liver reserve and function. Nutritional status was assessed using neo-GPS. GPS and neo-GPS values of ≥1 were defined as high scores [11]. Previously defined cutoff values of 40 [2] and 4.0 [4] were used for PNI and NLR, respectively. Values of 150–300 and >300 were used to define intermediate and high PLR values, respectively, as previously reported [5]. A CONUT score of 0–1 was defined as normal, whereas CONUT scores of 2–4 and ≥5 were defined as intermediate and high, respectively. Due to the lack of data on total cholesterol level and neutrophil count, CONUT score, and NLR could be determined in 215 and 269 patients, respectively.

### 2.4. Assessment of Tumor-Node-Metastasis Stage of HCC

In the present study, the sixth edition of the Tumor-Node-Metastasis staging by the Liver Cancer Study Group of Japan (TNM-LCSGJ) (Appendix A) was used for the evaluation of tumor burden [16].

### 2.5. Clinicopathologic Variables, HCC Treatment Algorithms, and Surgical Procedures

Prior to surgery, all patients underwent evaluation with conventional liver function tests, indocyanine green retention rate at 15 min (ICG-R15), and alpha-fetoprotein (AFP) level. The authors used the updated treatment algorithm for HCC, which included a combination of the five following factors: liver function reserve, extrahepatic metastasis, vascular invasion, tumor number, and tumor size [17]. In cases where hepatectomy was considered, the extent of liver damage was determined using ICG-R15 to reach a decision. The following new treatment algorithm was used. Patients with Child-Pugh class A/B liver function without extrahepatic metastasis or vascular invasion were recommended to receive one of the three following regimens. The first regimen was either surgical resection or radiofrequency ablation with no priority for up to three HCCs measuring ≤3 cm or surgical resection as first-line therapy for patients with solitary HCC regardless of the tumor size. The second regimen included surgical resection, which was recommended as first-line therapy, and transarterial chemoembolization as second-line therapy for patients with up to three HCCs measuring >3 cm. The third regimen included the combination of embolization, hepatectomy, hepatic arterial infusion chemotherapy, and molecular targeted therapy for patients with HCC accompanied by vascular invasion without extrahepatic metastasis. The treatment plan was selected on a case-by-case basis with consideration of the following factors: liver function, condition of the HCC, and extent of vascular invasion.

The Brisbane 2000 Terminology of Liver Anatomy and Resections proposed by Strasberg et al. was used to classify surgical procedures [18]. Anatomic resection was defined as resection of the tumor together with the related portal vein branches and corresponding hepatic territory. Anatomic resection was classified as hemihepatectomy (resection of half of the liver), extended hemihepatectomy (hemihepatectomy plus removal of additional contiguous segments), sectionectomy (resection of two Couinaud subsegments) [19], or segmentectomy (resection of one Couinaud subsegment). All other nonanatomic procedures were classified as limited resections. Limited resection was used to manage both peripheral and central tumors. Partial hepatectomy, which allows adequate surgical margins, was used to manage peripheral tumors and those with extrahepatic growth. Conversely, enucleation was used to manage central tumors near the hepatic hilum or major vessels due to the difficulty and risk associated with achieving adequate margins. All histological specimens were reviewed by a senior pathologist to confirm the final diagnosis.

### 2.6. Evaluation of Complications following Surgical Resection

Complications associated with surgical resection were evaluated using the Clavien-Dindo classification [20], and a Clavien-Dindo grade of ≥3 was considered to indicate a significant complication in the present study.

### 2.7. Statistical Analysis

In the present study, Fisher’s exact test and the Kaplan-Meier method with the log-rank test were used. Prognostic factors for progression-free survival (PFS) and overall survival (OS) were analyzed using the Cox proportional hazards model. *p* values of <0.05 were considered to indicate statistical significance. The discriminatory abilities of predictive ability for prognosis using the c-index. Easy-R ver. 1.53 (Saitama Medical Center, Jichi Medical University, Saitama, Japan) [21], a graphical user interface for R (The R Foundation for Statistical Computing, Vienna, Austria), was used to perform all statistical analyses.

## 3. Results

The cohort characteristics are summarized in Table 1. Briefly, the median age was 72 (interquartile range, 64–77) years, and 204 (75.3%) patients were male. The Child-Pugh scores were 5 and 6 in 229 (84.5%) and 42 (15.5%) patients, respectively. The TNM-LCSGJ stages were I, II, III, and IVa in 31 (11.4%), 191 (70.5%), 45 (16.6%), and 4 (1.5%) patients, respectively. Open and laparoscopic hepatectomy were performed in 138 (50.9%) and 133 (49.1%) patients, respectively.

### 3.1. Evaluation of OS

The median survival time (MST) was not evaluable (NE) (95% confidence interval [CI] not applicable [NA]) in patients with a neo-GPS of 0, 4.7 (95%CI 3.5–7.8) years in those with a neo-GPS of 1, and 2.8 (95%CI 0.6–7.6) years in those with a neo-GPS of 2 (*p* = 0.002; Figure 1A). Conversely, the MST was NE (95%CI 6.2 years–NA) in patients with a GPS of 0, 2.8 (95%CI 2.4–4.7) years in those with a GPS of 1, and 4.4 (95%CI 0.4–NA) years in those with a GPS of 2 (*p* = 0.002; Figure 1B). The analysis of the association between PNI and OS revealed that the MST was 7.8 (95%CI 5.9–NA) years in patients with a high PNI and 3.1 (95%CI 1.6–NA) years in those with a low PNI (*p* = 0.022; Figure 1C). The analysis of the association between the CONUT score and OS (*n* = 215) revealed that the MST was 7.8 (95%CI 5.7–NA) years in patents with a normal CONUT score, NE (4.1 years–NA) in those with an intermediate CONUT score, and 7.6 (95%CI 1.9–NA) years in those with a high CONUT score (*p* = 0.554; Figure 1D). The analysis of the association between NLR and OS (*n* = 269) indicated that the MST was 7.8 (95%CI 6.1–NA) years in patients with a low NLR and 3.9 (95%CI 2.7–NA) years in those with a high NLR (*p* = 0.031; Figure 1E). The analysis of the association between PLR and OS revealed that the MST was 7.1 (95%CI 5.6–NA) years in patients with a normal PLR, 7.6 (95%CI 4.6–NA) years in those with an intermediate PLR, and 2.8 (95%CI 0.4–NA) years in those with a high PLR (*p* = 0.801; Figure 1F). The c-indexes for the ability of neo-GPS, GPS, PNI, CONUT score, NLR, and PLR to predict OS were 0.571, 0.555, 0.524, 0.538, 0.533, and 0.507, respectively. Compared with a neo-GPS of 0, the HRs to predict OS were 1.618 (95%CI 1.105–2.368, *p* = 0.013) and 2.709 (95%CI 1.387–5.293, *p* = 0.004) for neo-GPSs of 1 and 2, respectively.

As shown in Table 2A, factors that were significantly associated with OS in univariate analysis were elevated AFP (>100 ng/mL; HR 2.333, 95%CI 1.579–3.446, *p* < 0.001), tumor size of ≥5 cm (HR 1.825, 95%CI 1.271–2.620, *p* = 0.001), multiple tumors (HR 1.686, 95%CI 1.094–2.600, *p* = 0.018), and neo-GPS of ≥1 (HR 1.723, 95%CI 1.164–2.549, *p* = 0.007). In the multivariate analysis, AFP of ≥100 ng/mL (HR 2.190, 95%CI 1.493–3.211, *p* < 0.001), tumor size of ≥5 cm (HR 1.508, 95%CI 1.037–2.193, *p* = 0.032), multiple tumors (HR 1.784, 95%CI 1.178–2.703, *p* = 0.006), and neo-GPS of ≥1 (HR 1.554, 95%CI 1.074–2.247, *p* = 0.019) were significantly associated with OS.

### 3.2. Evaluation of PFS

The median PFS (mPFS) was 3.2 (95%CI 2.1–4.2) years in patients with a neo-GPS of 0, 2.1 (95%CI: 1.3–2.5) years in those with a neo-GPS of 1, and 0.8 (95%CI 0.4–4.8) years in those with a neo-GPS of 2 (*p* = 0.009; Figure 2A). The mPFS was 3.0 (95%CI 2.2–3.5) years in patients with a GPS of 0, 1.0 (95%CI 0.5–2.0) years in those with a GPS of 1, and 3.0 (95%CI 0.4–NA) years in those with a GPS of 2 (*p* < 0.001; Figure 2B). The analysis of the association between PNI and PFS revealed that the mPFS was 2.3 (95%CI 2.0–3.3) years in patients with a high PNI and 2.3 (95%CI 0.8–2.6) years in those with a low PNI (*p* = 0.034; Figure 2C). The analysis of the association between CONUT score and PFS (*n* = 215) indicated that the mPFS was 3.3 (95%CI 1.9–4.2) years in patents with a normal CONUT score, 2.1 (1.5–3.9) years in those with an intermediate CONUT score, and 2.1 (95%CI 1.1–6.6) years in those with a high CONUT score (*p* = 0.426; Figure 2D). The analysis of the association between NLR and PFS (*n* = 269) revealed that the mPFS was 2.5 (95%CI 2.0–3.3) years in patients with a low NLR and 1.2 (95%CI 0.8–2.5) years in those with a high NLR (*p* = 0.065; Figure 2E). The analysis of the association between PLR and PFS (*n* = 271) indicated that the mPFS was 2.3 (95%CI 2.0–3.2) years in patients with a normal PLR, 2.3 (95%CI 1.1–4.6) years in those with an intermediate PLR, and 1.6 (95%CI 0.2–NA) years in those with a high PLR (*p* = 0.737; Figure 2F). The c-indexes for neo-GPS, GPS, PNI, CONUT score, NLR, and PLR in predicting PFS were 0.555, 0.545, 0.515, 0.531, 0.526, and 0.514, respectively. Compared with a neo-GPS of 0, the HRs to predict PFS were 1.428 (95%CI 1.052–1.938) (*p* = 0.022) and 2.029 (95%CI 1.140–3.612) for neo-GPSs of 1 and 2, respectively (*p* = 0.016).

Table 2B summarizes the factors that were significantly associated with PFS. Briefly, elevated AFP (≥100 ng/mL; HR 1616, 95%CI 1.198–2.180, *p* = 0.0016), multiple tumors (HR 1.453, 95%CI 1.037–2.035, *p* = 0.030), and neo-GPS of ≥1 (HR 1.546, 95%CI 1.164–2.054, *p* = 0.003) were significantly associated with PFS in univariate analysis. The multivariate analysis indicated that AFP ≥ 100 ng/mL (HR 1.743, 95%CI 1.325–2.292, *p* < 0.001), multiple tumors (HR 1.537, 95%CI 1.148–2.057, *p* = 0.004), and neo-GPS of ≥1 (HR 1.522, 95%CI 1.186–1.954, *p* = 0.001) were significantly associated with PFS.

### 3.3. Rates of High-Grade Clavien-Dindo Complications

Table 2 summarizes the rates of Clavien-Dindo grade ≥3 complications and their associations with the nutritional indices evaluated in the present study. Only high neo-GPS and GPS values could predict Clavien-Dindo grade ≥3 complications (*p* = 0.007 and *p* = 0.042, respectively).

## 4. Discussion

In the present multicenter retrospective study including 271 patients with HCC and Child-Pugh class A liver function who underwent R0 resection, the neo-GPS, a recently developed assessment tool for hepatic function based on the ALBI grade [14], exhibited the best c-index for predicting OS and PFS compared with the currently used nutritional assessment tools. Importantly, neo-GPS was a significant predictor of OS and PFS based on the multivariate analysis using the Cox proportional hazards model. In addition, neo-GPS was a better predictor of high-grade (≥3) Clavien-Dindo complications compared with the other nutritional tools evaluated in the present study. Overall, these results (in Table 3) suggest that neo-GPS might be considered an important prognostic predictive assessment tool in patients with HCC.

The liver damage classification, which includes ICG-R15, is considered to be more suitable than the Child-Pugh classification for prognostic prediction in patients undergoing surgical resection. On the other hand, the ALBI score not only exhibits a good relationship with ICG-R15 (r = 0.563, 95%CI 0.550–0.570, *p* < 0.0001) [22] but also functions as a more detailed assessment tool than the liver damage classification [23]. Moreover, the ALBI score can act as an indicator of nutritional status based on its demonstrated good relationship with PNI (r = −0.939, 95%CI −0.95 to −0.92, *p* < 0.001) [3]. In addition, CRP is an important biomarker for the prognosis, recurrence, and treatment response in adult solid tumors [24]. Therefore, it is not surprising that neo-GPS, which utilizes the ALBI grade and CRP, exhibited the best predictive ability not only for prognosis but also for high-grade (≥3) Clavien-Dindo complications.

The differences in clinical features between HCC and other malignancies should be considered in the interpretation of our findings because HCC develops frequently in patients with chronic liver disease, including cirrhosis, who often suffer from amino acid imbalance. In a recent study, which defined amino acid imbalance as a branched-chain amino acid/tyrosine ratio of ≤4.4, the authors reported that borderline amino acid imbalance indicated by an ALBI score of −2.588 was similar to the cutoff value for ALBI grade 1 (−2.600) [12]. Therefore, in patients with HCC, neo-GPS, which uses the ALBI grade instead of a serum albumin cutoff of 3.5 g/dL, might act as a more sensitive nutritional marker compared with other nutritional assessment tools. Although the Japanese clinical treatment guidelines for liver cirrhosis recommend that nutritional intervention should be considered in patients with chronic liver disease and a serum albumin level of ≤3.5 g/dL [25,26], earlier nutritional intervention with the inclusion of branched-chain amino acid before the development of hypoalbuminemia (≤3.5 g/dL) should be considered during the clinical course of chronic liver disease.

Our analyses indicated that neo-GPS exhibited not only a better prognostic predictive performance but also greater sensitivity for predicting risk of postoperative complications in patients undergoing hepatectomy for HCC compared with the other nutritional assessment tools that were examined in the current study, including GPS, PNI, NLR, PLR, and CONUT score. A higher rate of complications is expected in association with hepatectomy for HCC in patients with a high neo-GPS (≥1); therefore, nutritional intervention should be considered in these patients. The utility and significance of nutritional interventions in liver transplantation were reported by Kaido et al. [27]. A recent study reported that muscle volume decline was an important prognostic factor for recurrence and OS [28]. Another study reported that the frequency of muscle abnormalities increased with worsening hepatic reserve function [29]. In the present study, the authors did not elucidate the relationship between neo-GPS and muscle abnormalities due to the lack of information on muscle volume which precluded the determination of presarcopenia or sarcopenia. Moreover, despite the multicenter setting, the present study was limited by its retrospective design. Therefore, future studies should not only expand on the utility of neo-GPS as an assessment tool for prognostic prediction in HCC but also elucidate its association with sarcopenia.

## 5. Conclusions

In summary, the recently introduced neo-GPS based on the ALBI grade was a better prognostic nutritional assessment tool for the prediction of postoperative complications and prognosis in patients with HCC and Child-Pugh class A liver function undergoing surgical resection.

## Figures and Tables

**Figure 1 cancers-14-02091-f001:**
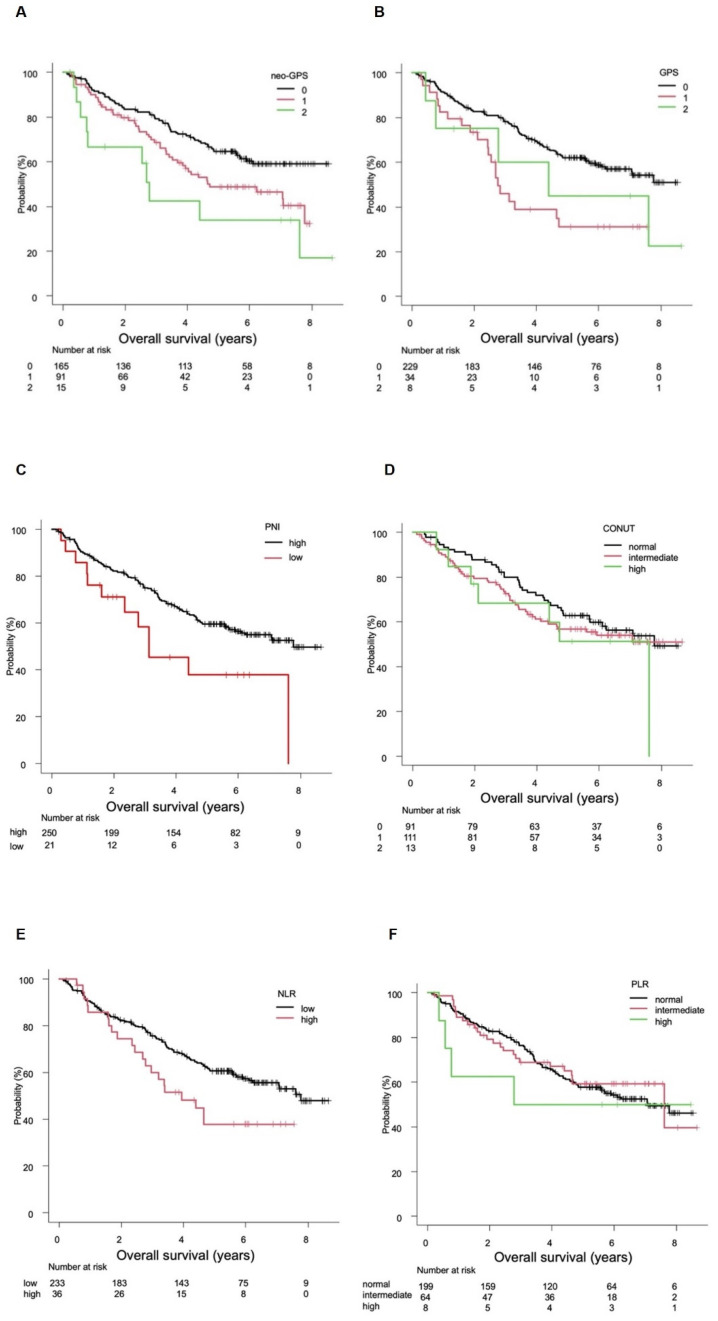
Overall survival (OS) according to assessment tool. (**A**) The median survival times (MSTs) were not evaluable (NE) (95% confidence interval [CI] not applicable [NA]–NA]), 4.7 (95%CI 3.5–7.8) years, and 2.8 (95%CI 0.6–7.6) years in patients with neo-Glasgow prognostic scores of 0, 1, and 2, respectively (*p* = 0.002). (**B**) The MSTs were NE (95%CI 6.2 years–NA), 2.8 (95%CI 2.4–4.7) years, and 4.4 (95%CI 0.4–NA) years in patients with GPSs of 0, 1, and 2, respectively (*p* = 0.002). (**C**) The MSTs were 7.8 (95%CI 5.9 years–NA) and 3.1 (95%CI 1.6–NA) years in patients with high (>40) and low (≤40) prognostic nutritional indexes (PNIs), respectively (*p* = 0.022). (**D**) The MSTs were 7.8 (95%CI 5.7–NA) years, NE (4.1 years–NA), and 7.6 (95%CI 1.9–NA) years in patients with normal, intermediate, and high controlling nutritional status scores, respectively (*p* = 0.554). (**E**) The analysis for neutrophil/lymphocyte ratio (NLR; *n* = 269) indicated that the MSTs were 7.8 (95%CI 6.1–NA) and 3.9 (95%CI 2.7–NA) years in patients with low (<4.0) and high (≥4) NLRs, respectively (*p* = 0.031). (**F**) The MSTs were 7.1 (95%CI 5.6–NA), 7.6 (95%CI 4.6–NA), and 2.8 (95%CI 0.4–NA) years in patients with normal, intermediate, and high platelet/lymphocyte ratios, respectively (*p* = 0.801).

**Figure 2 cancers-14-02091-f002:**
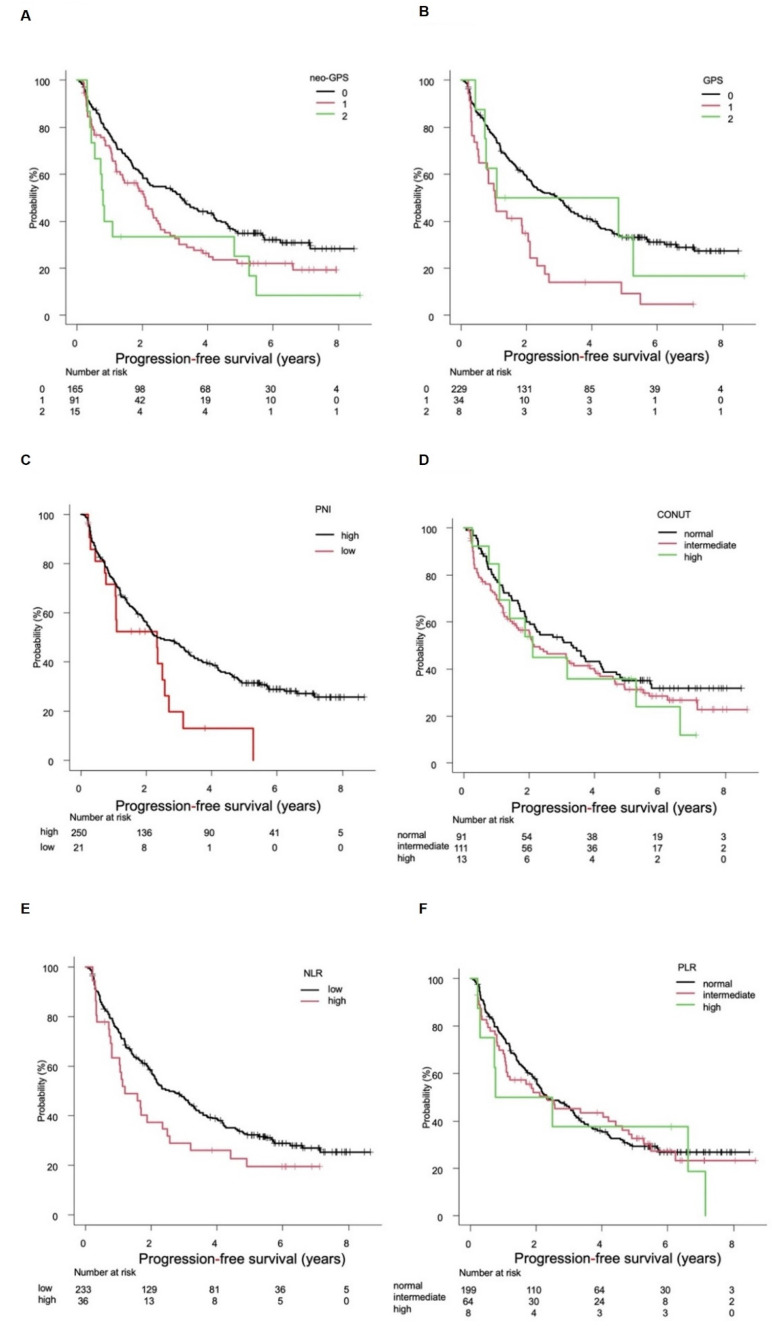
Progression-free survival (PFS) according to assessment tool. (**A**) The median PFS (mPFS) rates were 3.2 (95%CI 2.1–4.2), 2.1 (95%CI 1.3–2.5), and 0.8 (95%CI 0.4–4.8) years in patients with neo-GPS of 0, 1, and 2, respectively (*p* = 0.009). (**B**) The mPFSs were 3.0 (95%CI 2.2–3.5), 1.0 (95%CI 0.5–2.0), and 3.0 (95%CI 0.4–NA) years in patients with GPSs of 0, 1, and 2, respectively (*p* < 0.001). (**C**) The mPFSs were 2.3 (95%CI 2.0–3.3) and 2.3 (95%CI 0.8–2.6) years in patients with high (>40) and low (≤40) PNIs, respectively (*p* = 0.034). (**D**) The mPFSs were 3.3 (95%CI 1.9–4.2), 2.1 (95%CI 1.5–3.9), and 2.1 (95%CI 1.1–6.6) years in patents with normal, intermediate, and high controlling nutritional status scores, respectively (*p* = 0.426). (**E**) The mPFSs were 2.5 (95%CI 2.0–3.3) and 1.2 (95%CI 0.8–2.5) years in patients with low (<4.0) and high (≥4.0) neutrophil/lymphocyte ratios, respectively (*p* = 0.065). (**F**) The mPFSs were 2.3 (95%CI 2.0–3.2), 2.3 (95%CI 1.1–4.6), and 1.6 (95%CI 0.2–NA) years in patients with normal, intermediate, and high platelet/lymphocyte ratios, respectively (*p* = 0.737).

**Table 1 cancers-14-02091-t001:** Characteristics of patients (*n* = 271).

Patient Characteristics	
Age, years ***	72 (64–77)
Sex, male:female	204:67
Body mass index, kg/m^2^ *	23.2 (20.7–25.7)
Etiology, HCV:HBV:HBV&HCV:others	109:37:4:121
Positive for diabetes mellitus, *n* (%)	87 (32.1)
Aspartate transaminase, U/L *	36 (27–53)
Alanine aminotransferase, U/L *	31 (20–51)
Platelet count, 10^4^/µL*	15.9 (12.0–21.4)
Total bilirubin, mg/dL *	0.7 (0.5–0.9)
Albumin, g/dL *	4.1 (3.8–4.3)
Prothrombin time, % *	92.0 (83.5–100.7)
Child-Pugh score, 5:6	229:42
CRP, mg/dL *	0.09 (0.03–0.24)
ALBI score *	−2.77 (−2.48 to −3.02)
ICG-R15 (%) *	13.8 (9.5–18.8)
AFP, ng/mL *	12.3 (4.4–169.9)
Elevated AFP, ≥100 ng/mL, *n* (%)	75 (28.5%)
Single tumor, *n* (%)	223 (82.3)
Maximum tumor size, cm *	3.6 (2.5–6.0)
Resection, open:laparoscopic	138:133
Microvascular invasion present, *n* (%)	297 (69.2)
Macro-portal vein invasion, Vp2:Vp3	7:5
Macro-hepatic vein invasion, Vv2:Vv3	6:0
TNM-LCSGJ, I:II:II:IVa	31:191:45:4
Operation time, minutes *	334 (264–410)
Blood loss, mL *	550 (240–1094)
Observation period, years *	5.1 (2.4–6.3)
Death, *n* (%)	119 (43.9)
Neo-GPS, 0:1:2	165:91:15
Clavien-Dindo classification ≥3, *n* (%)	61 (22.5)

* Median. Values in parentheses show interquartile range, unless otherwise indicated. HCV, hepatitis C virus; HBV, hepatitis B virus; CRP, C-reactive protein; ALBI score, albumin-bilirubin score; ICG-R15, indocyanine green retention rate at 15 min; AFP, alpha-fetoprotein; GPS, Glasgow prognostic score; TNM-LCSGJ, Tumor-Node-Metastasis staging by the Liver Cancer Study Group of Japan 6th edition.

**Table 2 cancers-14-02091-t002:** Cox hazard analysis for overall survival and progression-free survival.

**(A) OS**	**HR**	**Univariate**	***p* Value**	**HR**	**Multivariate**	***p* Value**
		**95%CI**			**95%CI**	
Age, ≥75 years	1.217	0.828–1.787	0.318	-	-	-
Sex, female	0.683	0.428–1.090	0.110	-	-	-
Non-viral etiology	1.098	0.738–1.636	0.644	-	-	-
Positive for diabetes mellitus	1.086	0.723–1.632	0.690	-	-	-
Child-Pugh score, 6	0.825	0.474–1.436	0.496	-	-	-
Elevated AFP, ≥100 ng/mL	2.333	1.579–3.446	<0.001	2.190	1.493–3.211	<0.001
Tumor size, ≥5 cm	1.825	1.271–2.620	0.001	1.508	1.037–2.193	0.032
Tumor number, multiple	1.686	1.094–2.600	<0.001	1.784	1.178–2.703	0.006
Within Milan criteria	0.709	0.478–1.050	0.086	-	-	-
Laparoscopic resection	0.971	0.658–1.432	0.881	-	-	-
neo-GPS, ≥1	1.723	1.164–2.549	0.007	1.554	1.074–2.247	0.019
**(B) PFS**	**HR**	**Univariate**	***p* Value**	**HR**	**Multivariate**	***p* Value**
		**95%CI**			**95%CI**	
Age, ≥75 years	1.103	0.835–1.457	0.490	-	-	-
Sex, female	0.867	0.624–1.206	0.397	-	-	-
Non-viral etiology	1.207	0.918–1.587	0.178	-	-	-
Positive for diabetes mellitus	1.068	0.804–1.419	0.650	-	-	-
Child-Pugh score, 6	0.961	0.780–1.184	0.711	-	-	-
Elevated AFP, ≥100 ng/mL	1.616	1.198–2.180	0.002	1.743	1.325–2.292	<0.001
Tumor size, ≥5 cm	0.664	0.300–1.469	0.312	-	-	-
Tumor number, multiple	1.453	1.037–2.035	0.030	1.537	1.148–2.057	0.004
Within Milan criteria	0.456	0.208–1.003	0.051	-	-	-
Laparoscopic resection	0.820	0.627–1.073	0.149	-	-	-
neo-GPS, ≥1	1.546	1.164–2.054	0.003	1.522	1.186–1.954	<0.001

OS, overall survival; PFS, progression-free survival; HR, hazard ratio; 95%CI, 95% confidential interval; AFP, alpha-fetoprotein; GPS, Glasgow prognostic score.

**Table 3 cancers-14-02091-t003:** Rates of high-grade Clavien-Dindo complications according to the assessment tool.

Clavien-Dindo Complications			*p* Value
Neo-GPS	0	≥1	
Low-grade CD (<3)	137	73	
High-grade CD (≥3)	28	33	0.007
GPS	0	≥1	
Low-grade CD (<3)	183	27	
High-grade CD (≥3)	46	15	0.042
PNI	>40	≤40	
Low-grade CD (<3)	197	13	
High-grade CD (≥3)	53	8	0.100
CONUT score	Normal (≤1)	Elevated (≥2)	
Low-grade CD (<3)	73	93	
High-grade CD (≥3)	18	31	0.413
NLR	<4.0	≥4.0	
Low-grade CD (<3)	180	28	
High-grade CD (≥3)	53	8	1.000
PLR	<150	≥150	
Low-grade CD (<3)	156	54	
High-grade CD (≥3)	43	18	0.622

CD, Clavien-Dindo complication; GPS, Glasgow prognostic score; HCC, hepatocellular carcinoma; PNI, prognostic nutritional index; NLR, neutrophil/lymphocyte ratio; PLR, platelet/lymphocyte ratio, CONUT, controlling nutritional status.

## Data Availability

Due to the nature of this research, participants in this study could not be contacted regarding whether the findings could be shared publicly, thus supporting data are not available. The datasets generated and/or analyzed for the current study are not publicly available due to the nature of the research, as noted above.

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
