# Peer review of "Comparison of the New Neo-Glasgow Prognostic Score Based on the Albumin-Bilirubin Grade with Currently Used Nutritional Indices for Prognostic Prediction following Surgical Resection of Hepatocellular Carcinoma: A Multicenter Retrospective Study in Japan"

_cancers, 2022, doi:10.3390/cancers14092091_

Round 1

Reviewer 1 Report

In this study, the authors reported a retrospective study on the feasibility of using the neo-Glasgow prognostic score (GPS) as a nutritional prognostic assessment in 271 patients with HCC. The authors compared the prognostic predictivity of neo-GPS with several other established markers for nutritional assessment. It was concluded that neo-GPS was a good prognostic nutritional assessment tool for patients undergoing surgical HCC resection.

Line 130. It was mentioned that Kaplan–Meier method was used in the statistical analysis. The authors should include the Kaplan-Meier curves comparing the OS and PFS between patients with relatively high and low neo-GPS scores.

Author Response

Thank you for your useful comments, I sent two figures through journal’s WEB site, but it wasn't listed in the body of the text.

This time, I have posted figs. 1 and 2 in the text, could you please confirm these figures?

Reviewer 2 Report

Review report

The manuscript entitled “Comparison of the New Neo-Glasgow Prognostic Score Based on the Albumin-Bilirubin Grade with Currently Used Nutritional Indices for Prognostic Prediction Following Surgical Resection of Hepatocellular Carcinoma: A Multicenter Retrospective Study in Japan” by Masaki Kaibori is a comparative study on predictive and prognostic value of nutritional index, neutrophil/lymphocyte, and platelet/lymphocyte ratios. Based on their results, authors have shown that Neo-GPS which replaces albumin with albumin – bilirubin grade, was a good prognostic nutritional assessment tool for the prediction of postoperative complications and prognosis in patients undergoing surgical HCC resection. Even though, this  a retrospective analysis, this study is informative and I recommend that this manuscript may be published without any reservation.

A similar article has been published by the authors in Cancers (Kaibori, M., Hiraoka, A., Matsui, K., Matsushima, H., Kosaka, H., Yamamoto, H., Yamaguchi, T., Yoshida, K., & Sekimoto, M. (2022). Predicting Complications following Surgical Resection of Hepatocellular Carcinoma Using Newly Developed Neo-Glasgow Prognostic Score with ALBI Grade: Comparison of Open and Laparoscopic Surgery Cases. Cancers, 14(6), 1402. https://doi.org/10.3390/cancers14061402). This limits the novelty of the present article.  Nevertheless, the topic is relevant in the field and the prognostic score based on these studies may be able to stratify patients undergoing HCC resection in order to avoid post -surgical complications.

This study is different from earlier reports in the field, eg. Hiraoka, A et al 2017 who used albumin-bilirubin (T) grading system for the determination of hepatic function in HCC patients. 

Author Response

Thank you very much for your useful suggestion.
